# Distribution of myogenic stem cell activator, hepatocyte growth factor, in skeletal muscle extracellular matrix and effect of short-term disuse and reloading

So Kuwakado[1], Alaa Elgaabari[2,3], Kahona Zushi[2], Sakiho Tanaka[2], Miyumi Seki[2], Ryuki Kaneko[2], Yuji Matsuyoshi[2], Takahiro Suzuki[2], Kenichi Kawaguchi[1,4]*, Yasuharu Nakashima[1], Ryuichi Tatsumi[2]

1 Department of Orthopaedic Surgery, Faculty of Medical Sciences, Kyushu University, Fukuoka, Japan, 2 Department of Animal and Marine Bioresource Sciences, Graduate School of Agriculture, Kyushu University, Fukuoka, Japan, 3 Department of Physiology, Faculty of Veterinary Medicine, Kafrelsheikh University, Kafrelsheikh, Egypt, 4 Department of Rehabilitation Medicine, Kyushu University Hospital, Fukuoka, Japan

* kawaguchi.kenichi.241@m.kyushu-u.ac.jp

## Abstract

Hepatocyte growth factor (HGF) is a key myogenic stem cell (satellite cells) activator, that resides in the extracellular matrix (ECM). However, HGF distribution in the ECM varies depending on the muscle fiber type. Furthermore, aging impedes the binding of HGF to its receptors owing to nitration by peroxynitrite ($ONOO^-$). Though oxidative stress increases rapidly during muscle disuse atrophy, satellite cells are rapidly activated upon reloading. In this study, we investigated the distribution of HGF in the ECM in various muscle fiber types, and examined nitration of HGF in disuse and reloading models. Immunofluorescence staining was performed on the soleus (Sol), plantaris (Pla), and gastrocnemius (Gas) muscles of 10-week-old mice. Six mice were used to assess HGF distribution, while 12 mice, divided into control, disuse, and reloading groups were used for qualitative evaluation of nitrated HGF (nitroHGF). Student's t-tests and the Bonferroni correction were employed for statistical analysis ($p < 0.05/3 = 0.0167$). In Sol muscle, type IIa and IIx muscle fibers exhibited higher HGF distribution in the ECM ($61.5 \pm 1.0\%$ and $56.7 \pm 1.1\%$, respectively) than type I fibers ($32.3 \pm 1.0\%$; $p < 0.001$). In Pla and Gas muscle, type IIa $55.8 \pm 0.9\%$ and $58.8 \pm 1.5\%$, respectively) and type IIx fibers ($49.6 \pm 0.9\%$ and $48.9 \pm 1.1\%$, respectively) had significantly higher HGF distribution in the ECM than type IIb fibers ($18.6 \pm 0.9\%$ and $13.0 \pm 1.0\%$; $p < 0.001$, respectively). The amount of nitroHGF increased in the disuse group compared to that in the control group but decreased in the reloading group compared to that in the disuse group. This preferential HGF distribution around type IIa and IIx muscle fibers indicates a distinct mechanism for satellite cell activation, differing from the satellite cell-rich environment associated with

**Data availability statement:** All relevant data are within the paper and its Supporting information files.

**Funding:** This work was supported by Grants-in-Aid for Scientific Research (B) from the Japan Society for the Promotion of Science (JSPS KAKENHI, Grant Numbers 21H02347 and 24K01911) (all to Ryuichi Tatsumi). This research was supported in part by grant funds from the Uehara Memorial Foundation and the Ito Foundation (to Ryuichi Tatsumi). The funders had no role in the study design, data collection and analysis, decision to publish, or preparation of the manuscript and did not provide support in the form of salaries for any author.

**Competing interests:** The authors have declared that no competing interests exist.

type I fibers and the lower HGF association with type IIb fibers. Disuse-induced HGF nitration may inhibit satellite cell activation. Reloading likely triggers mechanisms that counteract nitration, enabling satellite cell reactivation in young muscle.

## Introduction

In the hypertrophy and atrophy of skeletal muscles, apart from the balance between muscle protein synthesis (MPS) and muscle protein breakdown (MPB) rates [1–3], muscle stem cells (satellite cells) repair damaged areas of the muscle and cause hypertrophy, thus providing new myonuclear cells. Satellite cells reside in the niche between the sarcolemma and the extracellular matrix (ECM) of the muscle fiber, and most of them are dormant [4,5]. Satellite cells are activated by generating an activation signal cascade through Ca channels that sense changes in skeletal muscles during exercise as mechanical stimulation. The activated satellite cells then migrate toward the injury site and re-enter the cell cycle from resting conditions to proliferate. At this stage, committed satellite cells are called myoblasts. After the proliferative phase, myoblasts exit the cell cycle and differentiate into mature myocytes. Finally, myocytes fuse together to form multinucleated myotubes and/or fuse to damaged myofibers, providing new myonuclear. At this stage, the expression of myosin heavy chain (MyHC) and other contractile proteins begins [5,6]. Although satellite cells are more abundant around slow-twitch muscle fibers, and myonuclear number increases predominantly in type I muscle fibers [7,8], resistance training preferentially induces increase in the number of myonuclear cells and hypertrophy in type II muscle fibers compared to type I muscle fibers [9]. In contrast, in clinical studies of older people who underwent habitual resistance training, type I muscle fibers showed increased myonuclear numbers, but type II muscle fibers did not [10]. This suggests that type II muscle fibers may have a different mechanism for controlling satellite cell activation than that of type I muscle fibers. As a possible candidate, we focused on the distribution of hepatocyte growth factor (HGF), a myogenic stem cell activator, in the ECM.

HGF was demonstrated to be unique in its ability to activate satellite cell division in vitro, at the fast in 1995 by Allen et al [11]. HGF is abundant in the ECM of uninjured muscle tissue [12,13] and is released rapidly from extracellular tethering when subjected to mechanical stretching in living muscle [14–17]. The activation signaling cascade of satellite cells includes a process where HGF binds to the cell membrane receptor c-met [6,13], and the ability to bind to the c-Met of HGF has been evaluated directly for its c-Met-binding activity by a sandwich ELISA-like assay in our study group. The internal layer of the ECM surrounding muscle fibers has two primary constituents, collagen type IV and laminin-2, which vary in distribution as a function of muscle fiber type [18]. Qualitative evaluation has shown that the distribution of HGF in the ECM is biased depending on the muscle fiber type [12,19]. We attempted to quantitatively evaluate the distribution of HGF around muscle fibers as the average signal intensity in immunofluorescence staining, but gave up owing to the difficulty in extracting the cross-sectional area of the extracellular matrix layer around a single

muscle fiber as a specified region and distinguishing it from the extracellular matrix of adjacent muscle fibers. To assess the distribution of HGF in the tissue surrounding myofibers, the activation rates of neuronal nitric oxide (NO) synthase (nNOS) [20] can be used as a reference. Sarcolemmal nNOS activity was assessed using modified NADPH-diaphorase (NADPH-d) histochemistry, and the length of NADPH-d-positive sarcolemma was calculated relative to the cross-sectional circumference (CSC). Therefore, the main objective of this study was to quantitatively evaluate the distribution of HGF in the ECM surrounding different types of myofibers using this method.

Furthermore, our research group discovered that HGF in the ECM was nitrated with aging and lost its ability to bind to the c-met [19,21]. In our past BrdU-activation assay experiment, 1-h low-frequent stretch cultures (LF: 5.0 cycles/min) activated satellite cells to a level equivalent to a positive control culture (2.5 ng/mL HGF), while a 1-h highly-frequent (HF: 8.6 cycles/min) stretch did not activate satellite cells. HGF has some tyrosine residue, which is aromatic amino acids, and two of the tyrosine residue (Y198, Y250) present on the binding surface of HGF with the receptor c-met [19]. Aromatic amino acids are easily nitrated by $ONOO^-$ formed from the major messenger NO and superoxide ($O_2^-$), and nitrotyrosine is a promising biomarker of oxidative stress caused [22]. In our past experiment, conditioned media from stretch cultures were evaluated for the nitration of tyrosine residues by western blotting, and nitrotyrosine was detected exclusively in 1-h HF stretch cultures. In vitro experiment of satellite cultures with nitroHGF, by acute exposure of HGF to $ONOO^-$ showed a decrease in the activation rate at the BrdU-activation assay and receptor c-met binding activity at c-Met binding assay (ELISA-like assay). From the above, nitroHGF shows poor binding possibly owing to a change in polarity, resulting in the failure to activate muscle satellite cells and repair of muscle damage sites [19,21]. Aging induces a chronic and gradual increase in oxidative stress, which contributes to sarcopenia [23–26]. Elgaabari et al. (2024) reported that nitroHGF accumulates in the ECM of skeletal muscles of mice and rats with age [19]. An effect of aging on skeletal muscle tissue is the inactivation of satellite cells. Englund et al. (2020) reported that young (5 months old) Pax7-diphtheria toxin A (DTA) mice treated with tamoxifen to deplete satellite cells showed slower growth in the long term [27]. Therefore, if the involvement of satellite cells induced by HGF in the ECM is important in human fast-twitch fibers, the accumulation of nitroHGF in the ECM is thought to be an essential etiology of sarcopenia.

Similar to the increase in oxidative stress associated with aging, oxidative stress also increases rapidly in disuse-induced muscle atrophy [28–31]. Therefore, increased nitroHGF likely inhibits satellite cell activation, even in disuse muscle atrophy. However, satellite cells are rapidly activated upon reloading after disuse [32]. Based on this, the second objective of this study was to investigate how disuse and reloading affect nitration of HGF and contribute to the characteristics of each muscle fiber type. This was assessed qualitatively using immunofluorescence staining.

## Materials and methods

### Ethical approval

Animal interventions were conducted in strict accordance with the recommendations of the Guidelines for Proper Conduct of Animal Experiments published by the Science Council of Japan and with ethical approval from the Kyushu University Institutional Review Board (A22-082-2).

### Animals

Six C57BL6/J mice (male, 9–11 weeks) were ordered from KBT Oriental Co Ltd, bred, and housed at 23 ± 2 °C and 55 ± 10% humidity on a 12 h light/dark cycle (lights on at 8 a.m.) with free access to regular food (CRF-1, Oriental Yeast Co., Ltd.; Tokyo, Japan) and water. After a week of recovery, the animals were euthanized and tissues were collected. The animals were euthanized in a non-fasted state with ad libitum access to food and moisture. The Sol, Pla, and Gas muscles of the left lower hind limb were dissected from each mouse, oriented in tissue OCT compound, and frozen in isopentane cooled with liquid nitrogen.

 

Twelve C57BL6/J mice (male, 9 weeks old) were obtained from KBT Oriental Co., Ltd. and bred under the same conditions as described above. After one week of recovery, mice were divided into three groups so that their mean body weights were as equal as possible: control group, disuse group, and reloading group, with four animals per group. After each group's intervention period ended, the animals were euthanized, and tissues were collected. After measuring body weight, the tibialis anterior muscle (TA), extensor digitorum longus muscle (EDL), Sol, and Gas of the right lower hind limb were individually dissected from each mouse, and the muscle weights were measured. The Sol, Pla, and Gas muscles of the left lower hind limb were dissected as a lump from each mouse, oriented in tissue OCT compound, and frozen in isopentane cooled with liquid nitrogen. Euthanasia was performed by isoflurane inhalation before cervical dislocation; all efforts were made to minimize suffering during the animal experiments.

## Hindlimb unloading

We used a disuse model of rodent hindlimb unloading [33] in which a strip of adhesive tape was attached to the tail of the mice. The tape was passed through a hole in the cage lid attached to a clip on the top of the cage. The animals could move around the cage, but their hind limbs could not touch the cage floor. Mice in the control group were unrestricted and allowed to move freely for 5 days. The disuse group mice were adapted to the disuse model and kept in a state of hind-limb unloading for 5 days. The reloading group mice were first adapted to the disuse model for 5 days and then allowed to spend the next 3 days free after removing the strip and releasing from hindlimb unloading. All mice were housed in a temperature-controlled room on a 12:12-h light-dark cycle with food pellets and transport agar (ORIENTAL YEAST Co., Ltd.) provided ad libitum.

## Direct-immunofluorescence microscopy

Some sets of serial cryo-sections (about 10-μm thickness) were prepared using a Leica CM1950 cryostat (Nussloch, Germany) and examined for muscle fiber type, distribution rate of HGF in ECM, short diameter of muscle fibers, or qualitative evaluation of HGF and nitroHGF by direct-immunofluorescence microscopy.

Muscle cross-sections were fixed with hot PBS and steam for 5 min and blocked with sterile donkey-serum solution (containing 2% normal serum, 1% BSA, 0.1% cold fish skin gelatin, 0.05% Tween 20, 0.01% avidin, 100 mM glycine, and 0.05% sodium azide in PBS, pH 7.2) for 1 h at 25°C, and stained with HiLyte Fluor™ 647-labeled anti- MyHC 1 monoclonal antibody (1:100 dilution), Alexa Fluor™ 350-labeled anti-MyHC 2A monoclonal antibody (1:100 dilution), HiLyte Fluorescein 500-labeled anti-MyHC 2X monoclonal antibody (1:200 dilution) and Alexa Fluor™ 594-labeled anti-MyHC 2B monoclonal antibody (1:200 dilution) overnight at 4°C. Then, other set of serial cryo-sections were fixed with hot PBS and steam for 5 min and blocked with sterile donkey-serum solution for 1 h at 25°C, and stained with HiLyte Fluor™ 647-labeled anti-MyHC 1 monoclonal antibody (1:100 dilution), Alexa Fluor™ 350-labeled anti-MyHC 2A monoclonal antibody (1:100 dilution), Alexa Fluor™ 594-labeled anti-MyHC 2B monoclonal antibody (1:200 dilution) and anti-laminin monoclonal antibody (Sigma-Aldrich, Burlington, MA, USA, L9393; 1:50 dilution) overnight at 4°C, and were then probed with a TRITC-conjugated goat anti-rabbit IgG secondary antibody (Thermo Fisher, Waltham, MA, USA, 1515529; 1:250 dilution) for 1 h at RT. In addition, another set of serial cryo-sections were fixed with citric acid buffer heated in microwave for 5 min, treated with TrueBlack (Biotium, Fremont, CA, USA) to suppress autofluorescence due to lipofuscin for 5 min and blocked with 3% BSA for 1 h at 25 °C before incubation overnight at 4 °C in Alexa Fluor 594-labeled anti-HGF monoclonal antibody (1:50 dilution in the sterile solution containing 1% BSA, 0.1% cold fish skin gelatin, 0.05% Tween 20, and 0.05% sodium azide in PBS) and anti-Laminin monoclonal antibody (Sigma-Aldrich, L9393; 1:50 dilution) overnight at 4°C, and were then probed with a TRITC-conjugated goat anti-rabbit IgG secondary antibody (Thermo Fisher, 1515529; 1:250 dilution) for 1 h at RT.

In next experiment, one set cross-sections were fixed with hot PBS and steam for 5 min and blocked with sterile donkey-serum solution for 1 h at 25°C, and stained with HiLyte Fluor™ 647-labeled anti-MyHC 1 monoclonal antibody (1:100 dilution), Alexa Fluor™ 350-labeled anti-MyHC 2A monoclonal antibody (1:100 dilution), HiLyte Fluorescein

500-labeled anti-MyHC 2X monoclonal antibody (1:200 dilution) and Alexa Fluor™ 594-labeled anti-MyHC 2B monoclonal antibody (1:200 dilution) overnight at 4°C. One other set of serial cryosections was fixed with citric acid buffer heated in a microwave for 5 min, treated with TrueBlack (Biotium) to suppress autofluorescence due to lipofuscin for 5 min, and blocked with 3% BSA for 1 h at 25°C before incubation overnight at 4°C in Alexa Fluor 594-labeled anti-HGF monoclonal antibody (1:50 dilution). Another set of serial cryo-sections were stained with Fluorescein 500-labeled anti-nitrated Y198 HGF monoclonal antibody (1:50 dilution) overnight at 4°C. These antibodies have been used in previous studies and their specificity has been confirmed [19,34].

Sections were mounted in VECTASHIELD Antifade Mounting Medium (Vector Lab., Burlingame, CA, USA) and observed under a THUNDER Imager Live Cell (Leica microsystems). In imaging, the background fluorescence of the negative control without antibody was corrected at a threshold that did not show the fluorescence

## Morphometric analyses

Micrographs were analyzed using morphometric measurements (ImageJ software). To measure each muscle fibers, we distinguished each muscle fiber using the ECM identified by laminin, and compared serial sections to randomly select 30 fibers for each muscle and muscle fiber type for each individual. However, certain individuals in Sol had fewer than 30 type IIx fibers; therefore, we measured the maximum number of fibers. Myofiber size was evaluated by measuring myofiber cross-sectional area (CSA), and the Minimal Feret's diameter as short diameter, as its value is independent of orientation allowing obliquely oriented myofibers. Subsequently, to calculate the distribution rates of HGF in ECM around each muscle fiber types, we measured the length of ECM, as the cross-sectional circumference, rebelled with anti-laminin; and the length of HGF distribution in ECM, rebelled with anti-HGF antibody. The antibody-positive areas at the boundaries between the muscle fibers being measured and adjacent muscle fibers were not distinguished and were all included in the distribution length.

## Statistical analyses

Student's t-tests were employed for statistical analysis of experimental results between two of the three groups and the Bonferroni correction to adjust for multiple comparisons, using Microsoft Excel X. Data are presented as mean ± standard error for 3 or 4 mice per group. We chose a significance threshold of $\alpha = 0.05/n$, where n represents the number of comparisons, to account for the increased risk of false positives owing to multiple testing. This correction method helps maintain the overall Type I error rate at an acceptable level while controlling for the familywise error rate. Finally, a P-value below 0.0167 (0.05/3) was considered statistically significant and was indicated throughout by (*). Results represent independent experiments.

## Results

### Distribution of HGF in ECM at each muscle fiber type

The muscle fiber type proportion was obtained by quadruple immunostaining for the different MyHC isoforms (n = 6). In soleus (Sol) muscle, type I muscle fibers accounted for 29.3%, type IIa muscle fibers 63.4%, and type IIx muscle fibers 7.2%. Plantaris (Pla) and gastrocnemius (Gas) muscles contained almost no type I muscle fibers, and type IIa muscle fibers were less abundant than in Sol muscle (19.8% and 6.1%). Type IIx muscle fibers were more abundant in Pla muscle (20.8%) and increased in number but only in a small proportion in the Gas muscle (6.4%). However, type IIb muscle fibers were absent in Sol muscle but accounted for 59.4% and 86.9% of Pla and Gas muscles (Fig 1a, S1 Table).

Next, we created co-stained images of MyHC I, IIa, IIb, and laminin, which are abundant proteins in the ECM, and of HGF and laminin from serial sections to visualize the distribution of HGF in the ECM surrounding each muscle fiber (Fig 1b, S2 Table). Thirty muscle fibers were randomly selected for each muscle fiber type from the stained images of each muscle,

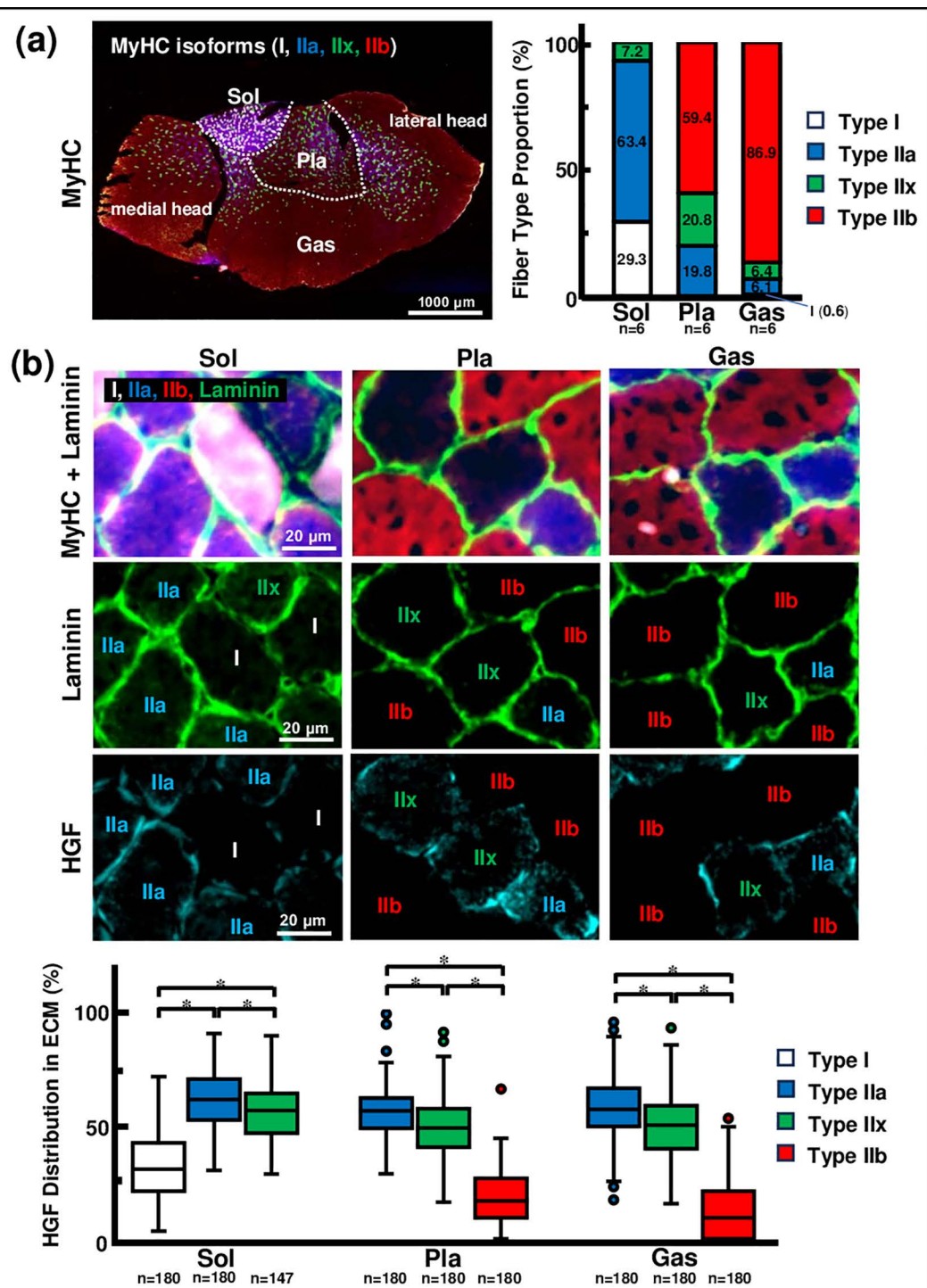

**Fig 1. The distribution of HGF in ECM around each muscle fiber types.** [a] Proportion of muscle fiber types in the Sol, Pla, and Gas muscles (n = 6). In rodents used in animal experiments, slow-twitch muscles are abundant in type I and type IIa muscle fibers and contain few type IIx muscle fibers. In contrast, fast-twitch muscles are abundant in type IIb muscle fibers, contain some type IIa and IIx muscle fibers, and have few type I muscle fibers. [b] HGF distribution in the ECM of each muscle fiber type in the Sol, Pla, and Gas muscles (n = 147–180). The length of the ECM surrounding each muscle fiber was determined using an anti-laminin antibody. Total HGF length in the ECM was determined using an anti-HGF antibody. In the Sol, IIa and IIx muscle fibers had a higher HGF distribution in the ECM than I muscle fibers (p < 0.001). In the Pla and Gas muscles, IIa and IIx muscle fibers

had a higher HGF distribution in the ECM than IIb muscle fibers (p<0.001). MyHC: Myosin heavy chain, Sol: Soleus muscle, Pla: Plantaris muscle, Gas: Gastrocnemius muscle, HGF: Hepatocyte growth factor. Statistically significant difference between the two groups at p<0.0167 (0.05/3) is indicated throughout by (*).

and the length of ECM surrounding each muscle fiber labeled with anti-laminin antibody or the HGF distribution on ECM labeled with anti-HGF antibody was measured using Image J; then, the distribution rate was calculated (n=147–180). Total HGF length in the ECM was determined using an anti-HGF antibody. In Sol muscle, type IIa, IIx muscle fibers (61.5±1.0%, 56.7±1.1%) have higher HGF distribution in ECM than type I muscle fibers (32.3±1.0%; p<0.001). In Pla muscle, IIa, IIx muscle fibers (55.8±0.9%, 49.6±0.9%) have higher HGF distribution in ECM than IIb muscle fibers (18.2±0.9%; p<0.001), and IIa muscle fibers have higher HGF distribution in ECM than IIx muscle fibers (p=0.006). Additionally, in Gas muscle, IIa, IIx muscle fibers (58.8±1.5%, 48.9±1.1%) have higher HGF distribution in ECM than IIb muscle fibers (13.0±1.0%; p<0.001), and IIa muscle fibers have higher HGF distribution in ECM than IIx muscle fibers (p=0.001).

### Changes in short diameter and proportion of muscle fiber type due to disuse and reloading model

In the second experiment, 12 wild-type mice were divided into three groups and housed for 5 or 8 days: a control group, a disuse group, and a reloading group (Fig 2a). In the disuse model, muscle atrophy is most rapid in type I muscle fibers with abundant mitochondria producing $O_2^-$ [28–31]. Therefore, $ONOO^-$ seems to be formed by $O_2^-$ reacting with NO more easily in slow-twitch muscles such as Sol [19,21]. In addition, type IIa and IIx muscle fibers with abundant HGF in the ECM constituted the majority of Sol. Therefore, in this experiment, we focused on the sol muscle in the disuse and reloading models. Short-term disuse did not induce significant atrophy in muscle weight/body weight ratio of Sol muscle (n=4) among control (1.00±0.03), disuse (0.97±0.06) and reloading group (0.98±0.03) (Fig 2b, S3 Table). Thirty muscle fibers were randomly selected for each muscle fiber type from the stained images of the Sol muscles in each group, then CSA and the Minimal Feret Diameter of each muscle fiber, as the short diameter, was measured using ImageJ (n=96–120). Regarding CSA, type I muscle fibers were larger in the control group (1360.5±43.4 $μm^2$) than in the disuse (1071.7±29.5 $μm^2$; p<0.001) and reloading groups (958.4±25.0 $μm^2$; p<0.001). Type IIa muscle fibers were also larger in the control group (1058.3±28.1 $μm^2$) than in the disuse group (949.0±27.8 $μm^2$; p=0.006) but did not differ from the reloading groups (984.4±35.7 $μm^2$). Type IIx muscle fibers were larger in the disuse group (1291.3±46.9 $μm^2$) than in the control group (1060.1±38.1 $μm^2$; p<0.001) and reloading group (1051.2±29.4 $μm^2$; p<0.001). Regarding the Minimal Feret Diameter, type I muscle fibers was longer in the control group (34.2±0.6 μm) than in the disuse (29.9±0.5 μm; p<0.001) and reloading groups (29.2±0.5 μm; p<0.001). Type IIa muscle fibers did not differ among the control (29.8±0.5 μm), disuse (28.1±0.5 μm), and reloading groups (28.7±0.6 μm). Type IIx muscle fibers were also did not differ among the control (29.2±0.7 μm), disuse (31.5±0.6 μm), and reloading groups (29.9±0.5 μm) (Fig 2c, S4 Table). In the proportion of muscle fiber types (n=4), type I muscle fibers did not differ among the control (35.5±4.7%), disuse (30.3±0.5%), and reloading groups (33.7±1.7%). Type IIa muscle fibers also did not differ among the control (56.6±5.2%), disuse (58.0±2.3%), and reloading groups (49.5±3.1%). Type IIx muscle fibers were larger in the reloading group (16.8±2.1%) than in the control group (7.9±0.6%; p=0.006), but not in the disuse group (11.7±2.3%) (Fig 2d, S5 Table).

### Signal intensity of HGF and nitroHGF due to disuse and reloading model

In the qualitative evaluation based on antibody signal intensity, the amount of nitro-HGF was more in the disuse group than that in the control group. In contrast, the amount of nitro-HGF was less in the reloading group than that in the disuse group (Figs 3a, S4).

Additionally, in the qualitative evaluation based on antibody signal intensity, the amount of HGF was more in the disuse group than in the control group and less in the reloading group than in the disuse group (Figs 3a, S4). To determine

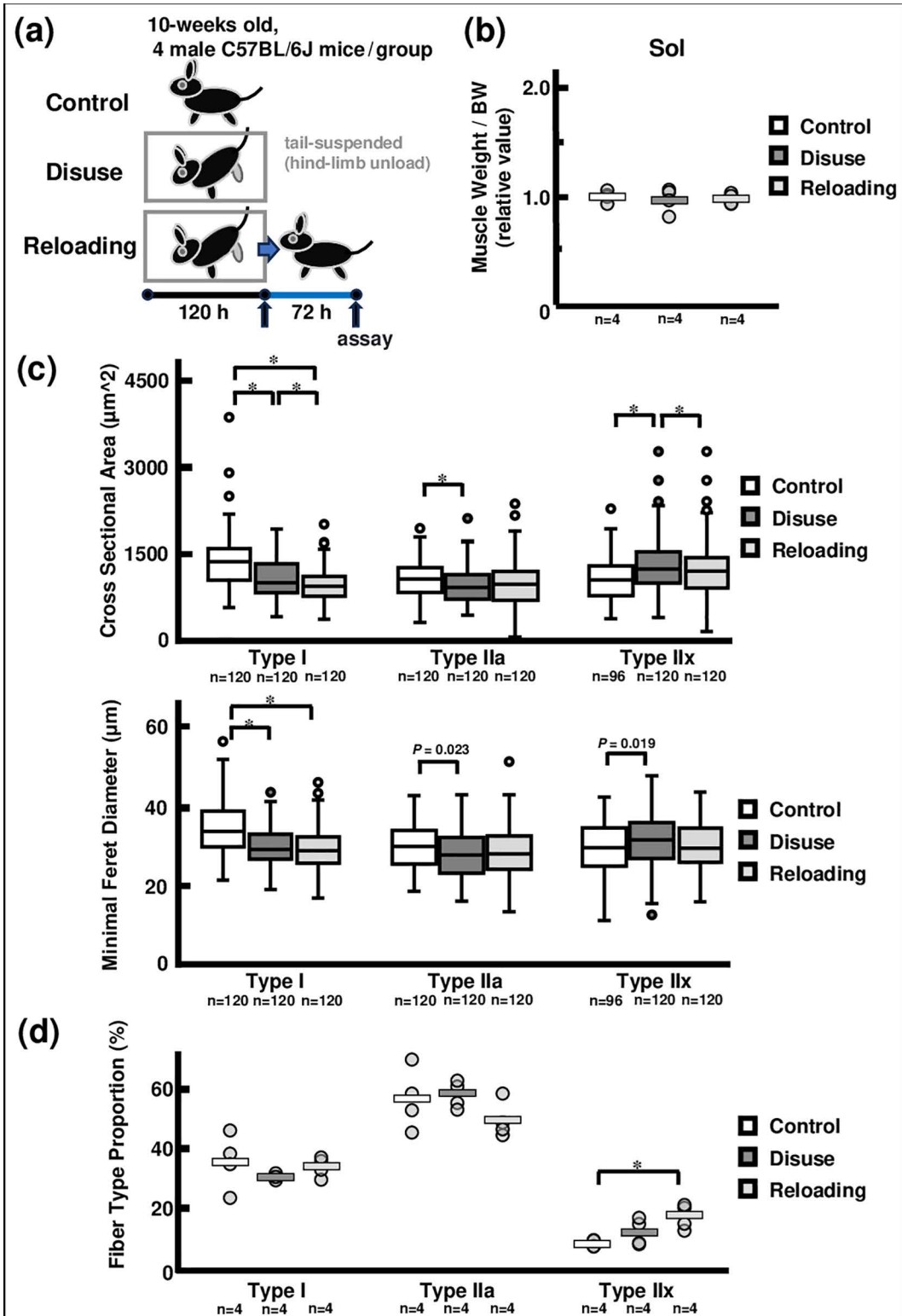

**Fig 2. Muscle fiber types in short-term disuse and short-term reloading model.** [a] Intervention schedule for the experiment. The mice in the control group (n=4) spent 5 days freely. The mice in the disuse group (n=4) were subjected to hind limb unloading for 5 days. The mice in the reloading

group (n = 4) were adapted to hind limb unloading for 5 days and allowed to spend the next 3 days freely. [b] Relative muscle weight to body weight in the control, disuse, and reloading groups (n = 4). The control, disuse, and reloading groups showed no significant differences. [c] Cross sectional area and minimal Feret diameter of each muscle fiber in the Sol (n = 96–120). Type I muscle fibers decreased in the disuse and reload groups compared to the control group (p < 0.001). [d] Proportions of muscle fiber types in the Sol for the control, disuse, and reloading groups (n = 4). Type IIx muscle fibers increased in the reloading group compared to those in the control group (p = 0.006). BW: Body weight. Statistically significant difference between the two groups at p < 0.0167 is indicated throughout by (*).

whether immobility due to the disuse model itself was the cause or whether increased oxidative stress was the cause, we focused on HGF around type IIa and type IIx muscle fibers in each muscle from the stained images of the control group, since slow-twitch muscles such as the Sol muscle have abundant type I muscle fibers that contain a large number of mito-chondria, which physiologically generate oxidative stress [35]. In the control group, the Sol muscle had more HGF in the ECM around type IIa and IIx muscle fibers than the Pla and Gas muscles (Figs 3b, S5).

## Discussion

The distribution of HGF in the ECM has not received as much attention as its distribution in satellite cells. This may be because type I muscle fibers, around which satellite cells are abundant [7,8], are more prevalent in slow-twitch muscles, whereas type IIb muscle fibers are the largest and most abundant in fast-twitch muscles (S1 and S2 Figs). However, since type I and IIa muscle fibers account for a high proportion of large mammals, such as humans, and type IIb is almost absent [36], medical research on humans should focus on HGF in the ECM surrounding type IIa and IIx muscle fibers as a characteristic of fast-twitch fibers and the activation of muscle satellite cells via the HGF/c-met pathway. Around IIb muscle fibers, satellite cells are less prevalent, and HGF is not widely distributed in the ECM. Therefore, type IIb muscle fibers may have mechanisms to uniquely enhance the self-repair capacity of muscle fibers [37] and MPS metabolism [1–3], which may not be mediated by satellite cell activation via the HGF/c-met pathway.

Many rodent studies that address myonuclear numbers have not focused on the distinction between type II muscle fibers, and few studies have shown clear benefits of the abundance of HGF in the ECM, which is characteristic of type IIa and IIx muscle fibers. A study on myonuclear number in humans reported that resistance exercise [36], especially unaccustomed exercise habits [38], increased the myonuclear number in type II muscle fibers more than in type I muscle fibers. In addition, a study focusing on mouse soleus muscles reported that type IIa and IIx muscle fibers hypertrophied more rapidly than type I muscle fibers in long-term reloading experiments after long-term disuse [39]. However, in our experimental system, the size of type IIx muscle fibers in the disuse group were larger than that of the control group, and the size of type IIx muscle fibers in the reloading group was smaller than that of the disuse group. In the disuse group, type IIx muscle fibers were significantly larger than type IIa muscle fibers, but this significant difference disappeared in the reload group. Furthermore, the proportion of type IIx muscle fibers in the reloading group were significantly higher than that of the control group (S3 Fig). As reported in other experimental studies of disuse muscle atrophy and spinal cord injury models [36,40], we hypothesize that this occurs through the transition into type IIx muscle fibers from other smaller muscle fibers. This transition may complicate the evaluation of the characteristic hypertrophy for each muscle fiber type. Further research to address the timing of muscle fiber type transition and innervation back to the original type [36] is needed to understand the unique relationship between satellite cells and each muscle fiber type.

In the short-term disuse model used in the second experiment, the muscle weight/body weight ratio of the Sol showed no significant change. However, type I muscle fibers were significantly smaller in the disuse group than in the control group, which we attributed to increased oxidative stress owing to the introduction of the disuse model, as reported in previous studies [28–31]. Therefore, a certain amount of $O_2^-$ was generated in our short-term disuse experiment, pos-sibly resulting in the formation of $ONOO^-$. Qualitative evaluation showed that nitroHGF increased in the skeletal muscle ECM of the disuse group. This result suggests that similar to aging, the activation of satellite cells is suppressed in disuse

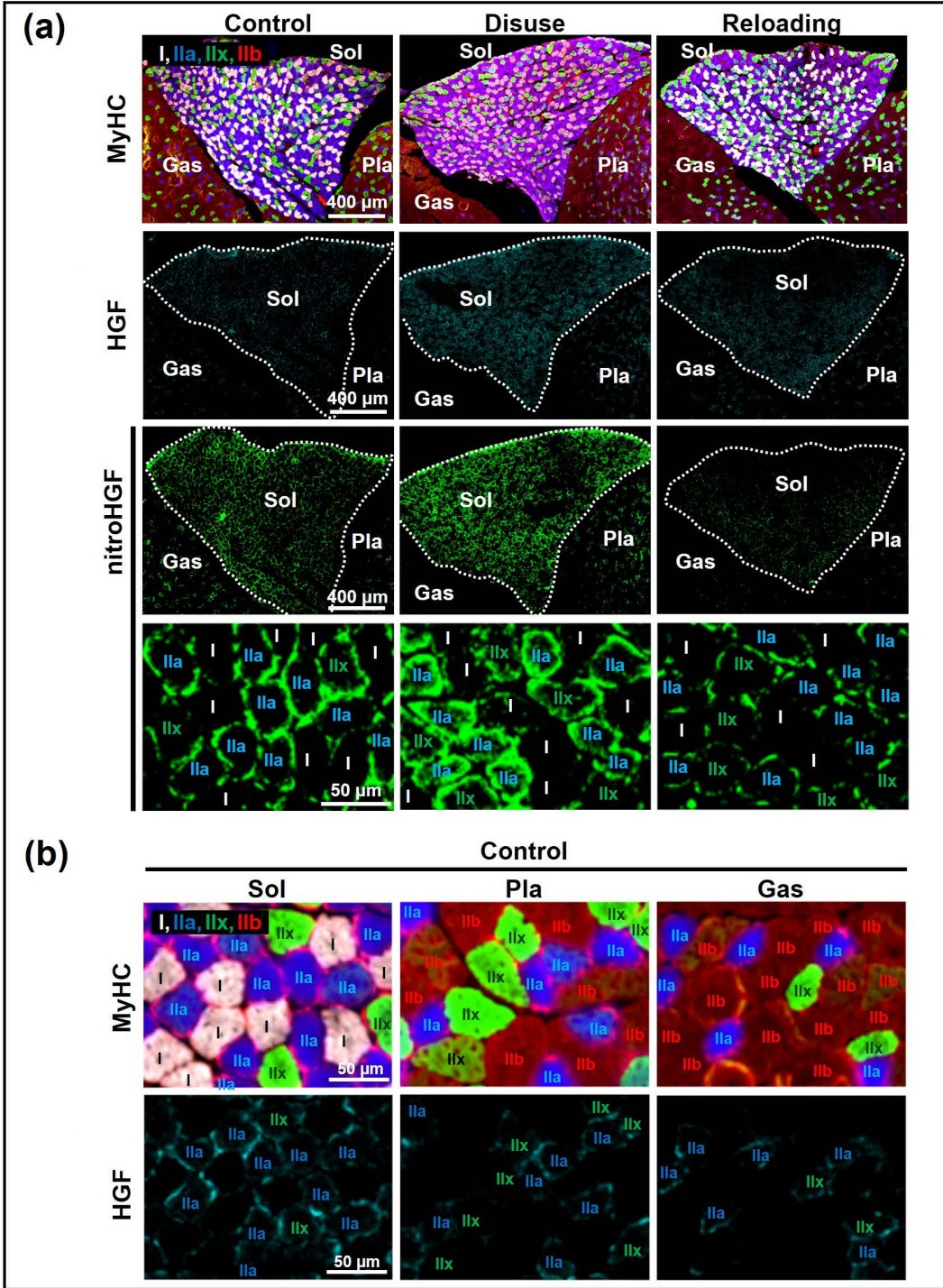

**Fig 3. Qualitative evaluation of nitration and increase in HGF levels in short-term disuse and reloading models by immunofluorescence staining.** [a] Qualitative evaluation of immunofluorescence staining with anti-HGF and anti-nitro-HGF antibodies. NitroHGF in the ECM was more in the disuse group than that in the control group and less in the reloading group than in the disuse group. In addition, HGF in the ECM was more in the disuse group than the control group and less in the reloading group than in the disuse group. [b] Qualitative evaluation of immunofluorescence staining with anti-HGF antibody in the control group. The amount of HGF around type IIa and IIx muscle fibers in the Sol was more than that in the Pla and Gas muscles.

muscle atrophy. In contrast, HGF and nitroHGF was less in the reloading group, so the release of HGF from the ECM did not appear to be inhibited. Although this is not a direct proof, this series of results can be explained by the reduction or decomposition of nitroHGF with reloading. However, since HGF reacts rapidly after being released by exercise from an ECM that has been anchored in advance and is not synthesized step-by-step [14–17], the decomposition of nitroHGF and resynthesis of HGF are unlikely. Therefore, we speculate that a reaction to reduce the nitration modification occurred. Because a rapid reaction occurs in muscle fibers due to ONOO$^-$ in the early stages of exercise [41] and exercise is repeated, researchers can reasonably assume that a mechanism reducing nitration modification may exist in muscle fibers. The mechanism, thereby assisting in the reactivation of satellite cells present in muscle fibers, may be like cytosolic enzymes such as CK [42] or LDH [43], which are released outside the cell when the sarcolemma is destroyed by exercise. This is a possible reason why the rapid hypertrophy of slow muscles is not suppressed when loading is resumed after disuse in young mice and rats [32].

Although disuse muscle atrophy is rapid [28–31], especially in young people, satellite cells are activated more rapidly upon reloading and contribute to the recovery of skeletal muscle mass than usual [32]. In the qualitative evaluation, HGF in the disuse group were more than those in the control group. This may be a mechanism that activates satellite cells more than usual, which is the reason for the rapid hypertrophy during reloading [32]. In the control group, even around the same type IIa and type IIx muscle fibers, the ECM of the Sol muscle appeared to contain more HGF than the Pla and Gas muscles. Slow-twitch muscles such as Sol are mostly made up of muscle fibers with many mitochondria, such as type I and type IIa muscle fibers [44]. Therefore, a larger amount of oxidative stress is generated than fast-twitch muscles such as Pla and Gas on a normal basis [35]. Although the detailed expression mechanism remains unknown, oxidative stress likely promotes HGF expression. In reloading after disuse, Sol muscle promoted the transition to type IIx muscle fibers, and this transition to type II muscle fibers is also induced by regular resistance exercise [9]. Few reports address resistance exercise. However, similar to aerobic exercise, an increase in oxidative stress has been reported [45]. In addition, Li, et al. (2024) reported the oxidative stress from endurance exercise promotes the transition of fast-twitch to slow-twitch muscle fibers, which contrasts with our findings [46]. It remains unclear whether aerobic or resistance exercise generates stronger oxidative stress. However, disuse muscle atrophy is associated with a substantial increase in reactive oxygen species, which causes rapid muscle atrophy [31]. Therefore, oxidative stress caused by disuse or exercise may play a role in modulating muscle fiber type transitions.

Our experiment has seven limitations: (1) The study was conducted on mice, and similar reactions in humans remain unguaranteed. Using human muscle tissue samples, confirming that HGF in the ECM is abundant around type IIa and IIx muscle fibers is necessary. (2) The merits of HGF distribution in the ECM have not been clearly characterized. The increase in myonuclear number and trends in muscle hypertrophy should be investigated by distinguishing between subtypes of type II muscle fibers. In particular, type IIb muscle fibers may hypertrophy faster than type IIa and IIx muscle fibers even if satellite cells are depleted in Pax7-diphtheria toxin A (DTA) mice treated with tamoxifen. (3) We used only nitrotyrosine as an oxidative stress marker, and the generation of reactive oxygen species was indirectly indicated by the atrophy of type I muscle fibers. A more detailed evaluation would require the use of other oxidative stress markers. (4) Other more accurate methods may exist for quantifying HGF levels in the ECM surrounding each muscle fiber. A means must be established to distinguish the extracellular matrix surrounding muscle fibers from adjacent muscle fibers and extract designated regions. (5) More quantitative methods are required to determine the degree of nitro-HGF. HGF is present in extremely low amounts in the skeletal muscle tissue, making it difficult to quantify by western blotting. However, a quantitative evaluation is necessary for a clear evaluation. (6) More detailed verification, such as reverse transcription polymerase chain reaction (PCR) is required regarding the increased expression of HGF in response to oxidative stress and the conditions for muscle fiber-type transition. (7) Insufficient normalization strategy for muscle fiber size. A normalization strategy that is not affected by muscle atrophy, such as some kind of bone length ratio, should have been used to normalize for size differences between individuals.

In conclusion, HGF may be abundant in the ECM around type IIa and IIx muscle fibers, providing new myonuclear cells and rendering them susceptible to muscle repair and hypertrophy by satellite cells via the HGF/c-Met pathway. This HGF distribution around type IIa and IIx muscle fibers appear to be a different control mechanism promote muscle hypertrophy from that of type I muscle fibers, which are surrounded by abundant satellite cells, and also distinguishes from type IIb muscle fibers. In disuse muscle atrophy, nitroHGF increases in response to oxidative stress. Satellite cells may be reactivated by a reduction mechanism for nitration during reloading, which has a rapid effect on type IIa and IIx muscle fibers in young individuals.

## Supporting information

**S1 Fig. Cross sectional area and minimal Feret diameter of each muscle fiber in the first experiment (n=147–180).** Regarding CSA, type IIx muscle fibers ($1144.9 \pm 27.7$ $\mu m^2$) were larger than type I ($1036.6 \pm 18.3$ $\mu m^2$; $p=0.001$) and IIa ($966.8 \pm 24.0$ $\mu m^2$; $p<0.001$) in Sol muscle. In Pla and Gas muscles, type IIb muscle fibers ($1695.4 \pm 46.6$ $\mu m^2$ and $1883.7 \pm 57.3$ $\mu m^2$) were larger than type IIa ($749.7 \pm 21.9$ $\mu m^2$ and $702.4 \pm 17.7$ $\mu m^2$; $p<0.001$) and IIx ($1064.2 \pm 25.1$ $\mu m^2$ and $888.0 \pm 25.0$ $\mu m^2$; $p<0.001$). Regarding the Minimal Feret diameter, type IIa muscle fibers ($28.3 \pm 0.4$ $\mu m$) are smaller than type I ($29.8 \pm 0.4$ $\mu m$; $p=0.010$) and IIx ($30.0 \pm 0.4$ $\mu m$; $p=0.007$), in Sol muscle. In Pla and Gas muscles, type IIb muscle fibers ($39.1 \pm 0.5$ $\mu m$ and $40.1 \pm 0.7$ $\mu m$) are larger than type IIa ($23.9 \pm 0.4$ $\mu m$ and $24.0 \pm 0.4$ $\mu m$; $p<0.001$) and IIx ($29.4 \pm 0.5$ $\mu m$ and $27.0 \pm 0.4$ $\mu m$; $p<0.001$). MyHC, myosin heavy chain; Sol: Soleus muscle; Pla: Plantaris muscle; Gas: Gastrocnemius muscle; HGF: Hepatocyte growth factor. Statistically significant differences between the two groups at $p<0.0167$, as indicated by (*).
(TIF)

**S2 Fig. Muscle fiber type proportions of Sol, Pla, and Gas muscles in the control group in the second experiment (n=4).** In Sol muscle, type I muscle fibers accounted for 33.2%, type IIa muscle fibers were 59.2%, and type IIx muscle fibers were 7.6%. In Pla muscle, type IIa muscle fibers accounted for 10.0%, type IIx muscle fibers for 13.5%, and type IIb muscle fibers for 76.5%. In Gas muscle, type I muscle fibers accounted for 0.6%, type IIa muscle fibers for 3.7%, type IIx muscle fibers for 4.5%, and type IIb muscle fibers for 91.2%.
(TIF)

**S3 Fig. Cross sectional area and minimum Feret diameter of each muscle fiber in Sol (n=96–120).** Regarding CSA, type I muscle fiber ($1360.5 \pm 43.4$ $\mu m^2$) were lager than IIa ($1058.3 \pm 28.1$ $\mu m^2$; $p<0.001$) and IIx muscle fibers ($1060.1 \pm 38.1$ $\mu m^2$; $p<0.001$) in the control group. In the disuse group, type IIx muscle fibers ($1283.0 \pm 47.8$ $\mu m^2$) were larger than type I ($1071.7 \pm 29.5$ $\mu m^2$; $p<0.001$) and IIa ($949.0 \pm 27.8$ $\mu m^2$; $p<0.001$). In the reloading group, type I ($958.4 \pm 25.0$ $\mu m^2$), IIa ($984.4 \pm 35.7$ $\mu m^2$), and IIx muscle fibers ($1051.2 \pm 29.4$ $\mu m^2$) showed no differences in size. Regarding the minimal Feret diameter, type I muscle fiber ($34.2 \pm 0.6$ $\mu m$) were lager than IIa ($29.8 \pm 0.5$ $\mu m$; $p<0.001$) and IIx muscle fibers ($29.2 \pm 0.7$ $\mu m$; $p<0.001$) in the control group. In the disuse group, type IIa muscle fibers ($28.1 \pm 0.5$ $\mu m$) were smaller than type I ($29.9 \pm 0.5$ $\mu m$; $p=0.011$) and IIx ($31.5 \pm 0.6$ $\mu m$; $p<0.001$). In the reloading group, type I ($29.2 \pm 0.5$ $\mu m$), IIa ($28.7 \pm 0.6$ $\mu m$), and IIx muscle fibers ($29.9 \pm 0.5$ $\mu m$) showed no differences in size.
(TIF)

**S4 Fig. Qualitative evaluation of immunofluorescence staining with anti-HGF and anti-nitro-HGF antibodies in other individuals.** NitroHGF in the ECM was more in the disuse group than that in the control group and less in the reloading group than in the disuse group. In addition, HGF in the ECM was more in the disuse group than in the control group and less in the reloading group than in the disuse group.
(TIF)

**S5 Fig. Qualitative evaluation of immunofluorescence staining with anti-HGF antibody in the control group.** The amount of HGF around type IIa and IIx muscle fibers in the Sol was more than that in the Pla and Gas muscles.
(TIF)

**S1 Table. Proportion of muscle fiber types in the Sol, Pla, and Gas muscles (n = 6).** This is the table of data in Fig 1a. Muscle fiber types were identified using co-staining images of anti-MyHC and anti-Laminin antibodies, and the number of each type was counted. Proportion was obtained for each skeletal muscle by dividing the number of each muscle fiber type by the total number of muscle fibers. Type: Muscle fiber type, Sol: Soleus muscle, Pla: Plantaris muscle, Gas: Gastrocnemius muscle, MyHC: Myosin heavy chain.
(TIF)

**S2 Table. HGF distribution in the ECM of each muscle fiber type in the Sol, Pla, and Gas muscles (n = 147–180).** This is the table of data in Fig 1b. The types of muscle fibers were identified using co-stained images of anti-MyHC and anti-Laminin antibodies, and were compared with co-stained images of anti-HGF and anti-Laminin antibodies in serial sections to randomly select 30 fibers for each muscle and muscle fiber type for each individual. Laminin (μm): The length of the ECM surrounding each muscle fiber was determined using an anti-laminin antibody. HGF (μm): Total HGF length in the ECM was determined using an anti-HGF antibody. Rate (%): Distribution rate of HGF in the ECM was calculated with dividing the total length of HGF by the length of the ECM. HGF: Hepatocyte growth factor, ECM: Extracellular matrix.
(DOCX)

**S3 Table. Relative muscle weight to body weight in the control, disuse, and reloading groups (n = 4).** This is the table of data in Fig 2b. Body weight and the weight of each skeletal muscle were measured for each individual, and the ratio was calculated. To calculate relative values, the ratio of skeletal muscle weight to body weight in the disuse and reloading groups was divided by the ratio of skeletal muscle weight to body weight in the control group. BW: Body weight.
(DOCX)

**S4 Table. Cross sectional area and minimal Feret diameter of each muscle fiber in the Sol (n = 96–120).** This is the table of data in Fig 2c. The types of muscle fibers were identified using co-stained images of anti-MyHC and Laminin antibodies to randomly select 30 fibers for each muscle and muscle fiber type for each individual, and myofiber size was evaluated by measuring myofiber cross-sectional area (CSA) and the Minimal Feret's diameter as short diameter, using morphometric measurements (ImageJ software).
(DOCX)

**S5 Table. Proportions of muscle fiber types in the Sol for the control, disuse, and reloading groups (n = 4).** This is the table of data in Fig 2d. Muscle fiber types were identified using co-staining images of anti-MyHC and anti-Laminin antibodies, and the number of each type was counted. Proportion was obtained for each skeletal muscle by dividing the number of each muscle fiber type by the total number of muscle fibers.
(DOCX)

## Acknowledgments

Special thanks to Ms. Akiko Sato (Kyushu University, Japan) for the technical assistance with animal care. The authors thank Muscle & Meat Sci. Lab, Department of Animal and Marine Bioresource Sciences, Graduate School of Agriculture for the chance of the experiments. The authors thank the Center for Advanced Equipment and Educational Support, Faculty of Agriculture, Kyushu University for the use of the fluorescence microscope. We would like to thank Editage (www.editage.jp) for English language editing.

## Author contributions

**Conceptualization:** Yuji Matsuyoshi, Ryuichi Tatsumi.

**Data curation:** So Kuwakado.

Formal analysis: So Kuwakado.

Funding acquisition: Ryuichi Tatsumi.

Investigation: So Kuwakado.

Methodology: Ryuki Kaneko, Yuji Matsuyoshi, Takahiro Suzuki, Ryuichi Tatsumi.

Project administration: Alaa Elgaabari, Takahiro Suzuki.

Resources: Kahona Zushi, Sakiho Tanaka, Miyumi Seki.

Supervision: Kenichi Kawaguchi, Yasuharu Nakashima.

Visualization: So Kuwakado.

Writing – original draft: So Kuwakado.

Writing – review & editing: Kenichi Kawaguchi.

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
