## [Decision Letter · Decision Letter 0]

17 May 2025

Dear Dr. Kawaguchi,

There are a number of important limitations that require a substantial revision, and the manuscript will be re-reviewed. For example, there is a very small sample size and the statistics require a more rigorous assemsment rather than  t-tests without correction for multiple comparisons despite analyzing many groups/types, Furthermore, several of the methods are described but others lack critical details and some measurements have not been adequately quantified. Furhtermore additional data are required to support the conclusions

We look forward to receiving your revised manuscript.

Kind regards,

Stephen E Alway, Ph.D.

Academic Editor

PLOS ONE

Journal Requirements:

Special thanks to Ms. Akiko Sato (Kyushu University, Japan) for the technical assistance with animal care. The authors thank Muscle & Meat Sci. Lab, Department of Animal and Marine Bioresource Sciences, Graduate School of Agriculture for the chance of the experiments. The authors thank the Center for Advanced Equipment and Educational Support, Faculty of Agriculture, Kyushu University for the use of the fluorescence microscope. This work was supported by Grants-in-Aid for Scientific Research (B) from the Japan Society for the Promotion of Science (JSPS KAKENHI, Grant Numbers 21H02347 and 24K01911) (all to R.T.). This research was supported in part by grant funds from the Uehara Memorial Foundation and the Ito Foundation (to R.T.). The funders had no role in the study design, data collection and analysis, decision to publish, or preparation of the manuscript and did not provide support in the form of salaries for any author. We would like to thank Editage (www.editage.jp) for English language editing.

This work was supported by Grants-in-Aid for Scientific Research (B) from the Japan Society for the Promotion of Science (JSPS KAKENHI, Grant Numbers 21H02347 and 24K01911) (all to Ryuichi Tatsumi). This research was supported in part by grant funds from the Uehara Memorial Foundation and the Ito Foundation (to Ryuichi Tatsumi). The funders had no role in the study design, data collection and analysis, decision to publish, or preparation of the manuscript and did not provide support in the form of salaries for any author.

4. Please amend the manuscript submission data (via Edit Submission) to include author Ryuki Kaneko.

5. Please amend your authorship list in your manuscript file to include author Ryuichi Kaneko.

6. Please amend your list of authors on the manuscript to ensure that each author is linked to an affiliation. Authors’ affiliations should reflect the institution where the work was done (if authors moved subsequently, you can also list the new affiliation stating “current affiliation:….” as necessary).

Reviewers' comments:

Reviewer's Responses to Questions

**Comments to the Author**

1. Is the manuscript technically sound, and do the data support the conclusions?

Reviewer #1: Yes

Reviewer #2: No

Reviewer #3: Partly

2. Has the statistical analysis been performed appropriately and rigorously?

Reviewer #1: Yes

Reviewer #2: No

Reviewer #3: I Don't Know

3. Have the authors made all data underlying the findings in their manuscript fully available?

Reviewer #1: Yes

Reviewer #2: Yes

Reviewer #3: Yes

4. Is the manuscript presented in an intelligible fashion and written in standard English?

Reviewer #1: Yes

Reviewer #2: Yes

Reviewer #3: Yes

Reviewer #1: One major problem is that the description of image analysis is inadequate and needs to be corrected.

1. Representation of the word "tamoxifen" (p7 line6)

The wording ‘administration of tamoxifen’ does not adequately describe the content of Englund's study and should be amended as follows. Also, Englund et al. (2020) used 5-month-old mice in their experiments, so young is ambiguous and the age in months should be specified.

Englund et al. (2020) reported that young mice treated with tamoxifen to inhibit satellite cells showed slower growth in the long term [26].

->

Englund et al. (2020) reported that young (5 month old) Pax7-diphtheria toxin A (DTA) mice treated with tamoxifen to deplete satellite cells showed slower growth in the long term [26].

Similarly, the meaning of (2) administering tamoxifen in the discussion (p.24 lines 7-9) is unclear: tamoxifen is a drug that kills satellite cells in Pax7-diphtheria toxin A (DTA) mice with diphtheria toxin. It is a drug that kills satellite cells in Pax7-diphtheria toxin A (DTA) mice by diphtheria toxin and does not affect satellite cells when administered to wild-type mice. The myonuclear number does not lead to a method to distinguish between types IIa, IIx and IIb.

2. Imaging analysis methods

The methods for calculating HGF distribution and muscle fibre diameter are unclear, with only a brief mention in the Methods (p12 lines13-16) or Results section, which is insufficient. With regard to HGF distribution, as myofibres are adjacent to each other, overlap should occur in the ECM stained with anti-laminin for each adjacent type. This should be clearly stated. In any case, an Image analysis section should be set up in Methods, which should state how the laminin staining sites were processed in the neighbouring fibre types, how the Minimal Feret diameter of the myofibres was measured (what application?), the number of myofibres analysed, the number of myofibres analysed, the number of fibres analysed, and the number of myofibres analysed. ), how many myofibres per individual were analysed, and should be detailed to the extent that other researchers can reproduce the results.

Reviewer #2: Major issues: (1) Very small sample size (only 3 or 4 mice per group), (2) Statistical approach is basic — just t-tests without correction for multiple comparisons despite analyzing many groups/types, (3) Some methods are described but lack critical details, (4) No true quantification of some key measurements, only qualitative staining.

Observations are reasonably reported, but some strong mechanistic claims are made ("nitration inhibits satellite cell activation" etc.) that are not directly proven by the data. Functional assays (like satellite cell activity) are needed to really conclude that. Limitations are acknowledged, but the conclusions still stretch beyond the evidence.

See attachment for full comments.

Reviewer #3: This is an automated report for PONE-D-25-12995. This report was solicited by the PLOS One editorial team and provided by ScreenIT.

ScreenIT is an independent group of scientists developing automated tools that analyze academic papers. A set of automated tools screened your submitted manuscript and provided the report below. Each tool was created by your academic colleagues with the goal of helping authors. The tools look for factors that are important for transparency, rigor and reproducibility, and we hope that the report might help you to improve reporting in your manuscript. Within the report you will find links to more information about the items that the tools check. These links include helpful papers, websites, or videos that explain why the item is important. While our screening tools aim to improve and maintain quality standards they may, on occasion, miss nuances specific to your study type or flag something incorrectly. Each tool has limitations that are described on the ScreenIT website. The tools screen the main file for the paper; they are not able to screen supplements stored in separate files. Please note that the Academic Editor had access to these comments while making a decision on your manuscript. The Academic Editor may ask that issues flagged in this report be addressed. If you would like to provide feedback on the ScreenIT tool, please email the team at ScreenIt@bih-charite.de. If you have questions or concerns about the review process, please contact the PLOS One office at plosone@plos.org.

**Do you want your identity to be public for this peer review?** For information about this choice, including consent withdrawal, please see our Privacy Policy

Reviewer #1: No

Reviewer #2: No

Reviewer #3: No

---

## [Author Response · Author response to Decision Letter 1]

5 Jul 2025

The author sincerely hopes that my manuscript will be accepted.

---

## [Decision Letter · Decision Letter 1]

23 Jul 2025

Distribution of myogenic stem cell activator, hepatocyte growth factor, in skeletal muscle extracellular matrix and effect of short-term disuse and reloading

PONE-D-25-12995R1

Dear Dr. Kawaguchi,

We’re pleased to inform you that your manuscript has been judged scientifically suitable for publication and will be formally accepted for publication once it meets all outstanding technical requirements.

Kind regards,

Masoud Rahmati

Academic Editor

PLOS ONE

Additional Editor Comments (optional):

Reviewers' comments:

Reviewer's Responses to Questions

**Comments to the Author**

Reviewer #1: All comments have been addressed

2. Is the manuscript technically sound, and do the data support the conclusions?

Reviewer #1: Yes

3. Has the statistical analysis been performed appropriately and rigorously?

Reviewer #1: Yes

4. Have the authors made all data underlying the findings in their manuscript fully available?

Reviewer #1: Yes

5. Is the manuscript presented in an intelligible fashion and written in standard English?

Reviewer #1: Yes

Reviewer #1: The revised manuscript addresses the previous concerns appropriately. The experimental design, statistical analysis, and interpretation of results are sound. The conclusions are supported by the data, and the limitations are clearly acknowledged. I recommend acceptance of this manuscript for publication.

**Do you want your identity to be public for this peer review?** For information about this choice, including consent withdrawal, please see our Privacy Policy

Reviewer #1: No

---

## [Editor Report · Acceptance letter]

PONE-D-25-12995R1

PLOS ONE

Dear Dr. Kawaguchi,

I'm pleased to inform you that your manuscript has been deemed suitable for publication in PLOS ONE. Congratulations! Your manuscript is now being handed over to our production team.

Kind regards,

on behalf of

Dr. Masoud Rahmati

Academic Editor

PLOS ONE